# The Permeability and Selectivity of the Polyamide Reverse Osmosis Membrane were Significantly Enhanced by PhSiCl_3_

**DOI:** 10.3390/membranes11020142

**Published:** 2021-02-18

**Authors:** Junjie Yu, Kaifeng Gu, Binbin Yang, Kaizhen Wang, Yong Zhou, Congjie Gao

**Affiliations:** Center for Membrane and Water Science & Technology, Zhejiang University of Technology, Hangzhou 310014, China; 2111801302@zjut.edu.cn (J.Y.); 1112001047@zjut.edu.cn (K.G.); 2111801288@zjut.edu.cn (B.Y.); 2111826019@zjut.edu.cn (K.W.); gaocj@zjut.edu.cn (C.G.)

**Keywords:** polyamide film, phenyltrichlorosilane, situ hydrolysis, ammonia hydrolysis

## Abstract

The work briefly introduces the nano-composite reverse osmosis (RO) membrane with more permeability and selective performance, and we adopted the phenyltrichlorosilane precursor with better chemical stability and greater spatial resistance. The phenyltrichlorosilane concentration was mainly discussed in this work. The in-situ hydrolysis of phenyltrichlorosilane and the occurrence of ammonia hydrolysis make it effectively incorporated into the polyamide film. The covalent bond and hydrogen bond of phenyltrichlorosilane and polyamide (PA) can be realized. The phenyl group can extend in the polyamide polymer network and give the film corresponding functions. There will be fewer non-selective defects between phenyltrichlorosilane and PA. Under the premise of maintaining the water-salt selectivity of the membrane, along with the increase of benzene trichlorosilane loading, the 300% pure water flux can be achieved and the desalination rate remains at 98.1–98.9%. This reverse osmosis (RO) is suitable for household water purification.

## 1. Introduction

In the past few years, water shortages and pollution problems are getting worse and membrane water treatment technology has been given more and more attention [1]. Reverse osmosis technology is a very efficient and environmentally-friendly way to obtain clean water. The membrane is the core of the entire separation process and its performance determines the application scope and operating cost of membrane water treatment technology. Interfacial polymerization is the main method for preparing polyamide reverse osmosis membranes. Preparation of new monomers for interfacial polymerization, control of reaction parameters, and membrane surface modification are the main methods in traditional research [1]. However, the bulk material and type of the membrane have not been updated. The development and application of new membrane materials requires significant cost and time, which is unfavorable for production.

The introduction of inorganic particles into the thin-film composite (TFC) membrane can effectively improve the membrane performance. The membrane obtained by this method is a thin-film nanocomposite (TFN) membrane that has been widely researched. As has been confirmed, the TFN membrane has superior performance (including antifouling ability, chlorine resistance, etc.) of both polymer and inorganic materials [2,3]. With technology development, the preparation and application of nanomaterials are becoming more extensive. The TFN membrane has a wide application prospect. Its excellent performance has led to the TFN membrane becoming a research hotspot.

There are many inorganic nanoparticles that can be added to the PA layer, and many works have done different attempts, such as mesoporous zeolite molecular sieve (Zeolite) [4,5], oxides (SiO_2_) [6,7], carbon-based materials [8,9], a metal organic framework (MOF) [10,11], and metal nanoparticles [12], etc., has aroused the interest of researchers. However, there are still several key problems that cannot be solved: (1) the problem of agglomeration and dispersion of nanoparticles; (2) hydrophilic nanoparticles can be well dispersed in water but it is difficult to evenly disperse them in the oil phase, which leads to a limited loading of nanoparticles in the PA layer; (3) nanoparticles are only encapsulated in PA skin by physical action, which can easily cause the nanoparticles to dissolve during interfacial polymerization and/or filtration or easily form non-selective interface defects, leading to selective performance decline. Therefore, in order to improve the nanomaterials dispersion in solution and the selective interfaces formation, complex chemical modification of nanoparticles is often required.

In fact, among all nanoparticles, silicon-based nanoparticles have excellent properties. The large amount of silanol groups (Si-OH) present on them can form strong interactions with polymer chains, such as hydrogen bonds and covalent bonds, thereby significantly improving the interface compatibility between silicon-based nanomaterials and polymers [13,14,15]. This has been extensively applied in the field of in-situ polymerization and sol-gel technology. The sol-gel reaction of silica particles is often formed by the hydrolysis and polymerization of chlorosilane and/or siloxy groups. On the nanometer scale, silicon-based nanocomposite materials are simple to prepare and are promising nanomaterials and easily form nanoscale dispersions with polymer components [16]. 

Stable nanoparticles can be obtained by in-situ generation, and in-situ sol-gel and in-situ reduction technologies are widely used. The formation of in-situ generated nanoparticles and polymer film is completed in the same system. The nanoparticle precursors are uniformly dispersed in the solution system to form a molecular-level mixture, which is beneficial to enhance the compatibility between the components and build a strong molecular-level interface force. The most widely used nanoparticle precursors are metal alkoxides, metal chloroalkoxides, metal salts, and complex metal alkoxides. By selecting suitable nanoparticle precursors and precise control of the nanoparticle formation process, a large number of hydrophilic groups can accelerate the hydrophilic molecules adsorption on the surface of nanoparticles and further promote the affinity of nanocomposites. This is also an effective way to improve the pollution resistance of the membrane surface [17,18].

In in-situ reduction technology, the choice of reducing agent will affect the formation of nanoparticles. Stronger reducing agents can promote the formation of smaller monodisperse nanoparticles. However, reducing agents can also cause secondary pollution, and it is still difficult to control the generation of larger particles. Considering environmental issues and human health, choosing environmentally compatible and friendly solvents, reducing agents, and coating agents is the main strategy for "green" nanoparticles [19,20].

Due to the dispersibility and distribution state of nanoparticles, and the difficulty of strong interaction between them and polyamide materials, it is easy to cause non-selective interface defects, leading to reduced selectivity of the reverse osmosis (RO) membrane. In this paper, we use the less active chlorosilane precursor phenyltrichlorosilane (C_6_H_5_SiCl_3_, PhSiCl_3_). Phenyl has a larger space volume. It is also more compatible with polyamide. It has relative hydrophobicity and low reactivity and stability. The phenyl group will provide a lower surface energy and a larger space volume for the hydrophobicity of the polyamide film. Due to the existence of these reactions, we can realize the covalent bond and hydrogen bond connection between PhSiCl_3_ and PA polymer. The phenyl group can extend in the polyamide polymer network and give the film the corresponding function. There will be fewer non-selective defects between PhSiCl_3_ and PA polymer.

## 2. Materials and Methods

### 2.1. Materials

Phenyltrichlorosilane (PhSiCl_3_, PTS, ≥98%, MACKLIN), m-phenylenediamine (MPD, 99.5%), Isopar G (ExxonMobil), 1,3,5-benzenetricarbonyl trichloride (TMC, 98%), camphor sulfonic acid (CSA, ≥99%), triethylamine (TEA, 99%), and chloroform (CHCl_3_, 98%) was obtained from Aladdin reagent platform. The polysulfone (PSF) membrane was provided by Huzhou Research Institute of Zhejiang University of technology.

### 2.2. Membrane Preparation

The PA membrane (TFC and TFN) were prepared by immersing PSF basement membrane in amine aqueous solution containing 185 mM (2.0%, g/100 mL) MPD and 172 mM (4.0 %) CSA at pH 10.0 for 5 min. Then, using filter paper to absorb excess droplets on the surface, we were careful not to scratch the membrane surface. We poured a mixture of 0.4 mM (0.01%) TMC and a certain amount of PTS (0.0, 0.1mM (0.002%), 0.2 mM (0.004%), 0.3 mM (0.006%), 0.4 mM (0.008%), and 0.5 mM (0.01%)) Isopar G onto the PSF bottom membrane, saturated with amine water. After reacting for 30 s, the organic phase solution was discarded. The PA layer was formed on the PSF membrane surface. Heating curing at 80 °C for 10 min was used as the subsequent optimization process. At last, we rinsed the membrane surface with deionized water to remove unreacted monomers and solvents and prepared PTS composite membranes with different contents. The membranes TFC, TFN-0.1, TFN-0.2, TFN-0.3, TFN-0.4, and TFN-0.5 were used to represent the synthesized membranes. The loading amounts corresponding to PTS were 0.0, 0.1, 0.2, 0.3, 0.4, and 0.5 mM, respectively.

Scheme 1 indicates the reaction scheme of benzenetricarbonyl trichloride/phenyltrichlorosilane (TMC/PTS) and m-phenylenediamine (MPD)/H_2_O.

### 2.3. Membrane Characterization

The PA layer morphology was obtained by a scanning electron microscope (SEM, SU8010, Hitachi, Japan). For the front side of the membrane, we used conductive double-sided tape to fix a small sample on the sample table for observing the image on the membrane surface; for the back side of the film, we transferred the separation layer to the silicon wafer and fixed the silicon wafer on the sample table with conductive double-sided tape. The details of the transfer process are discussed in the Appendix A. Before testing, we dried the samples at 25 °C for 24 h. An ion sputtering instrument (MC1000, Hitachi, Japan) was used to spray gold on the membrane surface and back with nano-platinum gold (<5 nm), which can overcome the charging effect. Finally, we placed the processed sample in the chamber of the instrument and scanned at an acceleration voltage of 5–15 kV.

The more detailed PA layer information may be obtained by TEM (TEM, HT7700 EXALENS, Hitachi, Japan) (the acceleration voltage for the test was 100 kV). For the membrane cross-section, the membrane sample was embedded in fresh white resin and treated in a vacuum oven (60 °C, 24 h) [21]. We used an ultramicrotome (EM UC7, Leica Microsystems, Wetzlar, Germany) to slice the cured resin containing the membrane sample to obtain 70–100 nm slices and transferred the slices to a copper mesh for further observation. For the surface projection of the film, we used (N,N-Dimethylformamide) DMF or CHCl_3_ to peel off the separation layer, transferred the separation layer to a copper mesh, and dried for 24 h at 25 °C for further observation.

The PA layer chemical structure can be analyzed by X-ray photoelectron spectroscopy (XPS, K-Alpha +, Thermo Scientific, Waltham, MA, USA). Test conditions: Al Kα (1487 eV) is the excitation source, which can carry emission voltage ≤ 12 kV and emission current ≤ 6 mA and vacuum degree ≤ 2 × 10^−7^ mbar. The scanning mode was constant analyzer energy (CAE) mode, the sampling hole size was 400 × 400 μm^2^. For a broad-spectrum scan, the pass energy was 200 eV with a 1.0 eV step length; for a narrow spectrum scan, the pass energy was 20 eV with a 0.05 eV step length [22]. All binding energies were charged with reference to 284.8 eV of C 1s for charge correction, and Casa XPS 2.3.19 and Thermo avantage v5.52 software were used for data processing and photoelectron peak analysis. Before element analysis, the membrane was rinsed in methanol for 15 min to remove unreacted monomers and impurities in the separation layer before testing and then dried at 25 °C for 24 h. 

The chemical composition of the membrane surface was analyzed by a Fourier transform infrared spectrometer (ATR-FTIR, IS 50, Thermo Nicolet, Shanghai, China) with total reflection component. For total reflection infrared spectrum, the penetration thickness of infrared light was less than 200 nm in the range of 2600~4000 cm^−1^, and in the range of less than 2000 cm^−1^, the penetration thickness of infrared light was greater than 300 nm. Considering that the separation layer was less than 200 nm, the separation layer was peeled and transferred to a stainless metal gasket and dried at room temperature for 24 h in the range of 4000~400 cm^−1^, placed the separation layer close to the detection crystal head, and scanned 128 times with a resolution of 4 cm^−1^. The scan result only includes the chemical information of the separation layer.

Thermal analysis was performed on the TGA Q5000 (TGA, TA Instruments, New Castle, DE, USA) and the DSC thermal analyzer. Prior to analysis, the composite film was first peeled off from the nonwoven fabric. A polyamide membrane with a polysulfone (PSF) supporting membrane was heated from 30 °C to 700 °C (10 °C/min of heating rate, 0.05 L/min nitrogen flow).

The membrane contact angle (CA) was measured by a video optical contact angle tester (OCA50AF, Dataphysics, Germany) with a high-speed camera function. The membrane was dried and attached to the glass plate. The 1 μL deionized water dripped gently on the membrane surface, using a camera to obtain a dynamic image of the droplet over time and using the SCA 20 software to fit the contact angle of the droplet when it first touched the membrane surface. The deionized water was replaced with methanol, glycerol, and diiodomethane to obtain the contact angles of liquids of different polarities on the membrane surface. The mean value of six parallel tests was taken as the test result.

The zeta potential (ζ) on the surface of the membrane sample was measured by a zeta potential analyzer (SurPASS, Anton Paar GmbH, Graz, Austria). Before the test, we soaked the membrane sample in 10^−3^ mol/L KCl solution for 24 h to adapt to the test environment. We cut the sample into 1 × 2 cm^2^ and fixed it on the measuring sample table with double-sided tape. We tested the context concentration 10^−3^ mol/L KCl solution and NaOH and HCl solutions with a concentration of 0.05 mol/L, which were used to adjust the pH value of the test solution. The pH titration range was 3.0~10.0.

### 2.4. Membrane Performance Evaluation

Test performance of the membrane at a pressure of 1 bar, 7 bar, 10 bar, 15.5 bar, and 55.2 bar. The crossflow velocity was 68 L/h, and the pre-pressure was 60 min. The pure water flux was tested at 1 bar. Then, the flux and rejection rate of NaCl solution were tested at different pressures and concentrations: 7 bar, 500 ppm; 10 bar, 1500 ppm; 15.5 bar, 2000 ppm; 55.2 bar, 32,000 ppm. The effective diameter of the test sample was 5 cm, and the effective area was 19.6 cm^2^. The permeate volume flux was calculated by the volume of liquid flowing out per unit time per unit area, and the calculation equation was Formula (1):J = V/(A × ∆t)(1)
where V is the volume of permeate volume, ∆t is the filtration time, and A is the membrane effective test area.

The rejection rate R of the membrane can be expressed by detecting the change of salt on both sides of the membrane, and its calculation equation is Formula (2):R = (1 − C_p_/C_f_) × 100%(2)
where C_p_ is the permeate concentration and C_f_ is the stock solution concentration. In dilute solution, the relationship between salt concentration and conductivity is approximately linear. We used electrical conductivity in our formula calculation.

## 3. Results and Discussion

### 3.1. Membrane Morphologies and Structures

The PSF base membrane has a smooth membrane surface without a special structure (SI, Appendix A). The PA layer formed a special fold structure. The SEM of Figure 1 compared the morphology of the TFC blank membrane and the TFN membrane. It can be seen that the TFC membrane presented a typical structure of polyamide with dense spherical particles. Small cavernous protrusions were arranged between the spheres. However, the modified TFN membranes had more and larger polyamide leaves, which were laid on the surface of the films without rough ridge valley processes. The distance between the leaves of the TFN membrane modified by PTS was also significant. This kind of large leaf protrusion with a far distance does not form many open pollution sites. During the formation of the TFN composite layer, PTS was connected to the polyamide network by hydrolysis and substitution of amino hydrogen (Figure 2). With the increase of space volume of PTS, the reaction rate of hydrolysis polycondensation significantly reduced, which is conducive to the formation of a relatively small siloxane polymer network.

After removing the PSF support membrane, the back of the PA layer was characterized and analyzed by SEM [23]. It can be concluded that the PA back pore was 10–35 nm (Figure 3). The back-pore diameter of the porous TFC membrane was obviously larger, reaching 10.8 nm. The membrane peak pore diameters were12.1 nm (TFN-0.1), 13.2 nm (TFN-0.2), 16.1 nm (TFN-0.3), 17.3 nm (TFN-0.4), and 18.9 nm (TFN-0.5), respectively. The aperture curve fitting process is shown in the Appendix A. The preparation parameters of TFN membrane were the same as the TFC membrane. The different TFN membranes were formed with different concentrations of chlorosilane. The change of pore size of backside of PTS was small, but it was still very significant. The pore size of the back surface was enlarged, which was due to the increase of PTS loading. These enlarged pore sizes accelerated the interfacial polymerization reaction [24]. The rougher ridge-valley structure appeared to be due to the enlarged two-phase interface [25].

The PA membrane has a rough structure of ridge-valley. The PA ridge height of the original blank TFC membrane was about 100 nm. The PA layer and supporting membrane were separated in the process of TEM sealing. Compared to TFC, TFN-0.2 membrane showed a bright PA layer and darker nanoparticles (Figure 4). The size of nanoparticles decreased significantly to less than 10 nm. We found the multi-level internal structure characteristic [15] through careful observation. The membrane had a dense base, about 10–20 nm. Both globular and ridged processes originated from the basement of this layer. Large protuberances extended outward and curled like leaves. In the process of polyamide leaf curling, a large number of cavities were embedded in the polyamide membrane. The interfacial polymerization process mainly occurred between the emulsified layer at the junction of the organic phase and the water phase. MPD molecules close to the emulsified layer quickly participated in the polymerization reaction. The reaction is slower for the MPD away from the emulsified layer. The difference in the reaction causes the protrusion [26]. PTS accelerated the diffusion of MPD and water. The subsequent dissolution diffusion reaction in the organic solution was greatly promoted. There were more inner and outer cavities in TFN membranes. These voids effectively promoted the transfer of water molecules and significantly increased the effective filtration area of the membrane. PTS promoted the formation of TFN membranes with this characteristic structure. The detailed TEM images have been added to the Appendix A.

### 3.2. Membrane Structural Properties

The hydrophilicity and charge of the PA membrane were changed due to the introduction of PTS. After hydrolysis polycondensation and ammonolysis of PTS, the membrane interfacial properties were significantly affected by the Si-O-Si, Si-N, -C_6_H_5_, and -OH groups. Since the increase of the hydrophilic hydroxyl group in the membrane, a small amount of PTS can improve the hydrophilicity of the membrane (Figure 5A). In the following membrane performance evaluation, the pH value of the feed liquid was 6.2 ± 0.2. From the change rule of zeta potential in Figure 5B, with 3–10 pH value, the TFN membrane loaded with PTS showed slightly higher electronegativity, but it did not show a certain regularity with the increasing load of phenyltrichlorosilane. The change of hydrophilicity and contact angle was not obvious. Therefore, the increase of the properties of the film was mainly due to the change of the microstructure.

The ATR-FTIR spectrum of the PTS modified PA membrane is shown in Figure 6. As a result of the polymerization of precursor and water, the Si-C, Si-O-Si, and Si-O-H bonds appear in the infrared spectrum. Under the asymmetric tension of the Si-O-Si group in the vertical (-Si-O-Si-Ph) direction of the siloxane bond [27], the 1083 cm^−1^ peak appeared as the PA membrane. The contraction vibration of amide bond was accompanied by the appearance of a characteristic peak (1656 cm^−1^, 1610 cm^−1^, and 1541 cm^−1^). It can be inferred that a thicker PA layer is formed according to the increase of the peak absorptivity. The stretching vibration of the O-H (3306 cm^−1^) bond came from CO-O-H and Si-O-H, which was caused by the hydrolysis of TMC and PTS. The chemical reaction between PTS and MPD resulted in the formation of Si-N bond (868 cm^−1^). It is remarkable that the absorption peak strength of Si-O and Si-N bonds increased with the increase of PTS loading. This indicates that PTS was successfully involved in the PA layer through a covalent bond.

By means of XPS elemental analysis, the presence of Si 2p in TFN-0.2 (Table 1) indicated that PTS was successfully embedded into the PA membrane. The Si 2p of TFN-0.2 membrane modified by PTS was significantly higher. In Figure 7, we fitted the narrow C 1s peak of the XPS spectrum. The spectrum of C1s can be divided into five kinds of carbon with different binding energies: (1) 284.6 eV, including C=C, C-H, C-C, and C-Si; (2) 285.3 eV, including C-CO; (3) 285.9 eV, including C-N; (4) 287.8 eV, including N-C=O; (5) and 288.4 eV, including O-C=O. At binding energy 284.6 eV, the carbon content increased from 50.5% to 57.2%. Increasing the loading of PTS lead to the decrease of the C-N bond ratio from 30.4% to 21.9%. The results showed that PTS increased the proportion of the main peak carbon on the front of the membrane and created more hydrophobic group phenyl on the surface. The C-N bond content is an important basis to evaluate the degree of crosslinking of PA polymer. After adding 0.1 mM of PTS, the crosslinking degree of polyamide was almost the same as that of the blank TFC membrane. On the contrary, the crosslinking degree of TFN-0.2 membrane also showed a downward trend. This may have caused interface defects of the membrane and degradation of membrane performance.

The thermal stability of TFC and TFN membranes was tested by TGA in nitrogen, the 5% weight loss temperature of the membrane was shown in Figure 8B, and the mass loss peak began in Figure 8C,D. The temperature rate T_max_ of the maximum mass loss was about 530 °C. The addition of PTS improved the heat resistance of the membrane. All the membranes showed two thermogravimetric peaks. The temperature range of the first weight loss peak was from 100 to 250 °C. The evaporation of the remaining water caused this phenomenon. The moisture loss of TFN membrane was higher than that of TFC membrane, which indicates that the moisture content of polyamide membrane was directly proportional to the PTS content. The second weight loss peak was about 530 °C, which was caused by the introduction of PTS in the interfacial polymerization process (Figure 2). Under the influence of hydrolyzed PTS, flexible molecular fragments increased and the weight loss temperature of polyamide was reduced to 530 °C. This type of polyamide molecule occupies the main body of the TFN PA layer. The final stage of degradation represents the further decomposition of aromatic polyamide (600~700 °C). The silicon in the sample can be degraded. Put another way, PTS did not contribute to carbon residues that had no contribution to the carbon residue of the sample after thermal degradation. At 700 °C, the coke yield of TFC film was the highest, at 35.4%, while the residual coke in TFN membrane decreased correspondingly. This was due to the siloxane segments in the PA network.

### 3.3. Membrane Performance

The PTS in the PA layer effectively changes the membrane separation and permeability (Figure 9). The permeation flux increases under high pressure. The pure water permeability of including the TFC membrane was poor (only 2.4 L·m^−2^·h^−1^·bar^−1^). A small amount of PTS can significantly improve the pure water permeability coefficient to 167% (relative to TFC), reaching 4.0 LMH/bar (TFN-0.1). As PhSiCl_3_ concentration increased, pure water permeation flux also continued to increase. For the improvement of pure water coefficient, the TFN-0.2 membrane increased to 288% (relative to TFC), reaching 6.9 LMH/bar; the TFN-0.5 membrane increased to 404% (relative to TFC), reaching 9.7 LMH/bar. The increase of pure water permeation flux mainly comes from the change of the structure and composition of the polyamide membrane, such as the increase of internal and external voids in the microstructure and the increase of hydrophilic hydroxyl groups. It can also be inferred that the PA polymer layer was changed, and the screening effect of the membrane pore was weakened by the intervention of PTS. The variation of salt flux was similar to that of pure water. TFN membrane flux was significantly higher than the flux of TFC membranes (Figure 9B–E). Especially at low salt concentration and pressure (7 bar, 500 ppm NaCl, Figure 9B), the TFN-0.2 membrane flux increased significantly to 267% (relative to TFC), reaching 40 LMH, while the NaCl retention also improved (TFC 98.5%; TFN-0.2 98.9%).

The in-situ hydrolysis polycondensation reaction of PTS and the occurrence of the ammonolysis reaction made PTS effectively incorporated into the PA layer. They formed strong interactions with PA polymers through covalent bonds and hydrogen bonds. In the in-situ polymerized polyamide membrane of PTS, not only abundant hydroxyl and amide groups were produced, but also functional groups, such as phenyl, Si-O, and Si-N. By adjusting the amount of PTS added, the hydrophilicity and permeability of TFN were effectively improved.

High water permeability is a prominent advantage of the TFN membrane loaded with PTS, suitable for a household RO membrane and an ultra-low pressure RO membrane. For the high concentration salt system, compared with the TFC membrane, the selectivity of the TFN-0.1 membrane was slightly increased, but the TFN-0.2 decreased slightly. Therefore, when the PTS concentration is appropriate, the high selectivity and permeability can be obtained simultaneously. A small amount of PTS, such as in the TFN-0.2 membrane, slightly reduced the NaCl rejection rate by only 98.1–98.9%. The PhSiCl_3_ functionalized TFN-0.2 membrane increased the water flux rate by 300% (relative to TFC), while the salt retention rate was (98.1–98.9%). Considering compatibility and dispersion, adding PTS to the organic phase is an effective way to improve membrane performance. The above test and analysis have demonstrated this. For the low-pressure desalination process, the amount of PTS added can be adjusted to prepare high-performance TFN membranes.

## 4. Conclusions

The in-situ hydrolysis polycondensation and aminolysis reaction of PTS make PTS effectively incorporated into PA membrane, forming strong interactions between covalent bonds and hydrogen bonds with PA polymers. In the in-situ polymerization of PTS, the polyamide membrane not only produced abundant hydroxyl and amide groups, but also produced functional groups, such as phenyl, Si-O, Si-N, and so on. PTS adds holes on the back of the polyamide and voids in the cross section. Larger leaf-like structures are created on the top surface that are increasingly loose. The distance between the peaks of the polyamide blades becomes larger, which significantly reduces the location of valleys that are easily contaminated. In addition, PTS significantly improves the membrane permeability. With the PTS loading increasing, the water permeability coefficient effectively improved (from 2.4 LMH/bar to 6.9 LMH/bar) and the desalination rate was maintained at 98.1–98.9%, which was suitable for household and ultra-low pressure RO membranes. With an appropriate increase in PTS loading, the hydrophobic TFN membrane had a higher water-salt selectivity.

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
