# Peer review of "The Permeability and Selectivity of the Polyamide Reverse Osmosis Membrane were Significantly Enhanced by PhSiCl3"

_membranes, 2021, doi:10.3390/membranes11020142_

Round 1

Reviewer 1 Report

Manuscript Junjie Yu et al "The Permeability and Selectivity of Polyamide Reverse Osmosis Membrane Were Significantly Enhanced by PhSiCl3" currently not recommended for publication. In my opinion, the presented work should be revised in terms of scientific style, language, supplementing scientific discussion, analyzing the data obtained. The main disadvantage of the work is the small number of conclusions from the results obtained and their correlation with each other.

What are the mechanical properties of the membranes obtained?

Line 35. What does TFN mean?
Lines 254-259. The number of TEM photos can be reduced. It is also recommended to improve their quality in the same way as many described by the authors elements are difficult to distinguish.
Lines 268-270. "From the change rule of zeta potential of Figure 4 (B) with 3-10 pH value, the TFN membrane loaded with PTS shows slightly higher electronegativity "you must specify the corresponding pH range for which this is observed.
Line 318. What does The second weight loss peak (530 ℃) correspond to?
Lines 322, 323. Why At 700 ℃, the coke yield of TFC film is the highest, which is 35.4%, while the residual coke in TFN film decreases correspondingly?
Line 331. It is necessary to check the dimension "L · m-2 · L-1 · bar-1"
Line 356. "The membranes water / salt selectivity:" must be replaced with "The membranes transport properties (or separation and permeability): "
Section 3.3. Membrane Performance. This section is a listing of experimental results and requires revision to include mechanisms.
Line 342. Need to rephrase - "Especially at low salt concentration and low concentration"
Line 355. Figure 8 E-D. "water flux" should be replaced by "salt flux".

Author Response

Dear editor, reviewers:

Thank you for giving us the opportunity to submit a revised version of the manuscript. We would like to take this opportunity to express our gratitude to the editors and reviewers. Thank you for your valuable time and comments to help us improve the quality of the manuscript. In the revised manuscript, we try our best to respond to the comments made by the reviewers. We adopted the comments of the reviewer and made the following changes to the contents. Our reply manuscript and revised manuscript are marked in red.

For the initial submission, due to our negligence, we failed to provide the correct author email address. We sincerely apologize for it. Now attach the correct author email.

The following is the correct author email: Junjie Yu, [email protected]; Kaifeng Gu, [email protected]; Binbin Yang, [email protected]; Kaizhen Wang, [email protected]

Response to Reviewer 1 Comments

Manuscript Junjie Yu et al "The Permeability and Selectivity of Polyamide Reverse Osmosis Membrane Were Significantly Enhanced by PhSiCl3" currently not recommended for publication. In my opinion, the presented work should be revised in terms of scientific style, language, supplementing scientific discussion, analyzing the data obtained. The main disadvantage of the work is the small number of conclusions from the results obtained and their correlation with each other.

What are the mechanical properties of the membranes obtained?

Reply:

When we dissolve the PSF base membrane to obtain the PA skin layer for infrared, electron microscopy, and XPS characterization, the PA layer of the TFN membrane is easier to transfer to the silicon wafer. Doped nanoparticles are beneficial to improve the mechanical properties of the membrane. Therefore, compared to TFC membrane, the mechanical properties of TFN membrane have been improved.

Compared with TFC membrane, the mechanical properties of the prepared TFN membrane have been enhanced.

Line 35. What does TFN mean?

Reply:

TFN is a thin-film nanocomposite membrane. It is obtained by introducing nanoparticles into the thin-film composite membrane (TFC). It has been corrected in the manuscript (Section 1, the second paragraph, line 1~3).

Lines 254-259. The number of TEM photos can be reduced. It is also recommended to improve their quality in the same way as many described by the authors elements are difficult to distinguish.

Reply:

We have tried our best to make improvements. The unnecessary images have been moved to the supplementary information (Section 1). The following corrections have been made.

The original manuscript (Section 3.1, the third paragraph, line 3): We can see that with the increase of the concentration of PTS, the protuberance of TFN membrane is higher and higher, and the cross-section structure is more and more clear. This can be attributed to the introduction of silicon element and cavity, which increases the contrast of cross-section TEM image. In Figure 3 (C), the TFN0.2 film shows a bright PA layer and darker nanoparticles.

The modified manuscript: Compared to TFC, TFN-0.2 membrane shows a bright PA layer and darker nanoparticles (Figure 4).

The original manuscript (Section 3.1, the third paragraph, line 14): These voids will effectively promote the transfer of water molecules and significantly increase the effective filtration area of the membrane.

The modified manuscript: These voids will effectively promote the transfer of water molecules and significantly increase the effective filtration area of the membrane. PTS promotes the formation of TFN membranes with this characteristic structure. The detailed TEM images have been added to the supplementary information (Figure S 1).

Lines 268-270. "From the change rule of zeta potential of Figure 4 (B) with 3-10 pH value, the TFN membrane loaded with PTS shows slightly higher electronegativity "you must specify the corresponding pH range for which this is observed.

Reply:

In the membrane performance test, the pH of the feed liquid was 6.2±0.2. Under this condition, the negative charge of TFN membrane is higher. Corresponding instructions have been made in the manuscript (Section 3.2, the first paragraph, line 5).

Line 318. What does The second weight loss peak (530 ℃) correspond to?

Reply:

Figure 2. Interfacial polymerization process.

The hydrolyzed PTS is introduced into the polyamide molecular chain through a covalent bond (Figure 2), which results in a weight loss peak at 530 ℃. The thermal degradation stage of aromatic polyamide is 600~700 ℃. Under the influence of hydrolyzed PTS, flexible molecular fragments increase, and the weight loss temperature of polyamide is reduced to 530 ℃. This type of polyamide molecule occupies the main body of the TFN PA layer. So 530 ℃ is also the maximum weight loss temperature. The infrared spectroscopy (Figure 5) proved that the reaction process took place.

The related discussions have been made in the manuscript (Section 3.2, the fourth paragraph, line 8).

Lines 322, 323. Why At 700 ℃, the coke yield of TFC film is the highest, which is 35.4%, while the residual coke in TFN film decreases correspondingly?

Reply:

At 700 ℃, the PA layer has been completely decomposed. Due to the introduction of phenylsiloxane fragments in the TFN membrane, the percentage of carbon content decreases accordingly. Compared to TFC membrane, the coke yield of TFN membrane is lower.

Line 331. It is necessary to check the dimension "L · m-2 · L-1 · bar-1"

Reply:

This is our negligence. “L · m-2 · L-1 · bar-1” has been changed to “L · m-2 · h-1 · bar-1”.

Line 356. "The membranes water / salt selectivity:" must be replaced with "The membranes transport properties (or separation and permeability): "

Reply:

We sincerely accepted this suggestion and made the correction in the manuscript.

Section 3.3. Membrane Performance. This section is a listing of experimental results and requires revision to include mechanisms.

Reply:

The following discussion has been added to the manuscript (Section 3.3, the second paragraph).

The in-situ hydrolysis polycondensation reaction of PTS and the occurrence of the ammonolysis reaction make PTS effectively incorporated into the PA layer. They form strong interactions with PA polymers through covalent bonds and hydrogen bonds. In the in-situ polymerized polyamide membrane of PTS, not only abundant hydroxyl and amide groups are produced, but also functional groups such as phenyl, Si-O, and Si-N are produced. By adjusting the amount of PTS added, the hydrophilicity and permeability of TFN are effectively improved.

Line 342. Need to rephrase - "Especially at low salt concentration and low concentration"

Reply:

The expression has been replaced.

The original manuscript (Section 3.3, line 14): Especially at low salt concentration and low concentration

The modified manuscript: Especially at low salt concentration and pressure

Line 355. Figure 8 E-D. "water flux" should be replaced by "salt flux".

Reply:

This problem has been corrected.

Reviewer 2 Report

The presented approach to modification of RO membranes seems sound. I have several notes for improvement of presentation. First, to ease the understanding of occurring processes the authors are advised to present the schemes of chemical reactions and scheme of formation of «holes» that are described in text. Second, the authors are asked to check for spliced sentences, such as line 47's «so it is difficult to High loading is formed in the polyamide film», hence "must be improved" in both description of methods and presentation of the results; I've listed several of such splices below but might've missed some. Third, the authors are advised to check the introductions of abbreviations and add the missing ones. And finally and more subjectively, I had trouble with English in this article at times, and while it can be just me (not a native speaker and not even an expert in English myself), the authors might consult with an expert in English just to be safe.

Specific points:

Mechanical errors: line 35, «mixed matrix membrane (TFN) membrane»; line 47, «so it is difficult to High loading is formed in the polyamide film»; lines 72-73, «Water and pollution resistance [17-18].» (was it occasionally separated from the previous sentence?); at lines 93-95 some reagents start with a capital letter and some with small ones; line 94, «Phenyltrichlorosilane (PhSiCl3, PTS, ≥98%, MACKLIN) M-phenylenediamine» - missing comma?; lines 120-121, «and use conductive double-sided tape to hold the silicon wafer Fix it on the sample table»; lines 122-124, «Use an ion sputtering instrument (MC1000, Hitachi, Japan) on the surface, back and The section is sprayed with nano-platinum gold (<5 nm) to overcome the charging effect»; line 138, «vacuum degree ≤ 2 × 10−7 mbar» (missing superscript), line 141, «284.8eV» (missing space), lines 168-170, «The zeta potential (ζ) on the surface of the membrane sample was measured by a zeta potential analyzer (SurPASS, Anton Paar GmbH, Austria) was used to measure the zeta potential (ζ) on the surface of the membrane sample.»; lines 179-183, «Firstly, test the membrane permeation flux of pure water at 1 bar, and then test the concentration of 7 bar concentration 500 ppm, 10 bar concentration 1500 ppm, 15.5 bar concentration 2000 ppm, 55.2 bar concentration 32000 ppm concentration 2000 ppm Flux and retention of NaCl solution.» (where the last part came from?), line 278 «the Si-C Si-O-Si» (missing comma). I might've missed some, the authors are advised to check the typesetting of entire manuscript.

Lines 159,162 – I suggest to use the capital L for litres so they can't be mistaken for capital L. It is strange that some parts of the manuscript use capital L and some small l.

Line 35, «mixed matrix membrane (TFN)», and line 100, «PA membrane (TFC and TFN)» - I guess TFC means «thin film composite» and TFN means «thin film nanocomposite», but these abbreviations were not introduced.

Section 2.2 seems to be partly written in imperative. Can't say this is frequent stylistic choice. The authors might consider rewording.

Also on section 2.2: why only the concentrations and not amounts of substances are given?

Line 97, PSF abbreviation was not introduced.

Line 120 – «for the back side of the film, transfer the separation layer to the silicon wafer,» - can authors please provide details of transfer protocol?

Lines 168-176, regarding zeta potential protocol: besides formatting errors being fixed I would ask for a more detailed description. Did the authors use one sample or double samples? How the titration was implemented? Why NaOH and not KOH was used to adjust the pH of KCl solution?

Line 199, «cavernous protrusions» - maybe it was intended as «cavernous depressions»?

Line 204, «the reaction rate of hydrolysis polycondensation» - this reaction is not described in text. In fact, no reactions were shown. May the authors please provide the equations for mentioned reactions?

Figure 1 – did the authors compare SEM visualisations of modified membrane with SEM of pristine PSF substrate membrane?

Line 218 – I don't know what junction saving is. Maybe it's just me, but the authors might consider adding the elaboration.

What is the meaning of peak pore diameter (line 219)? Judging by given numbers and figured out its not a maximum or diameter. Is it mean value of pore diameters?

Figure 2: how the fitting curve (red) of pore diameters war built?

Lines 248-249, «We speculate that the protrusion is caused by the violent heterogeneous eruption of MPD in the water phase of PSF membrane pore [26].» - was porosity of PSF membrane evaluated?

Lines 332 onwards – I guess the percentages show us the increase of values in relation to membrane without the dopant? The authors may consider explicitly stating it.

Line 350, «A slight excess of PhSiCl3» - I don't understand the use of the word «excess» here. Why this amount of dopant was considered excessive?

Lines 353-354, «the adjustment space of the formula is increased, which is conducive to the stable control of industrial production.» - might be elaborated further, since I didn't see the discussion of adjustment space prior to this.

Author Response

Dear editor, reviewers:

Thank you for giving us the opportunity to submit a revised version of the manuscript. We would like to take this opportunity to express our gratitude to the editors and reviewers. Thank you for your valuable time and comments to help us improve the quality of the manuscript. In the revised manuscript, we try our best to respond to the comments made by the reviewers. We adopted the comments of the reviewer and made the following changes to the contents. Our reply manuscript and revised manuscript are marked in red.

For the initial submission, due to our negligence, we failed to provide the correct author email address. We sincerely apologize for it. Now attach the correct author email.

The following is the correct author email: Junjie Yu, [email protected]; Kaifeng Gu, [email protected]; Binbin Yang, [email protected]; Kaizhen Wang, [email protected]

Response to Reviewer 2 Comments

The presented approach to modification of RO membranes seems sound. I have several notes for improvement of presentation. First, to ease the understanding of occurring processes the authors are advised to present the schemes of chemical reactions and scheme of formation of «holes» that are described in text. Second, the authors are asked to check for spliced sentences, such as line 47's «so it is difficult to High loading is formed in the polyamide film», hence "must be improved" in both description of methods and presentation of the results; I've listed several of such splices below but might've missed some. Third, the authors are advised to check the introductions of abbreviations and add the missing ones. And finally and more subjectively, I had trouble with English in this article at times, and while it can be just me (not a native speaker and not even an expert in English myself), the authors might consult with an expert in English just to be safe.

Specific points:

Mechanical errors: line 35, «mixed matrix membrane (TFN) membrane»; line 47, «so it is difficult to High loading is formed in the polyamide film»; lines 72-73, «Water and pollution resistance [17-18].» (was it occasionally separated from the previous sentence?); at lines 93-95 some reagents start with a capital letter and some with small ones; line 94, «Phenyltrichlorosilane (PhSiCl3, PTS, ≥98%, MACKLIN) M-phenylenediamine» - missing comma?; lines 120-121, «and use conductive double-sided tape to hold the silicon wafer Fix it on the sample table»; lines 122-124, «Use an ion sputtering instrument (MC1000, Hitachi, Japan) on the surface, back and The section is sprayed with nano-platinum gold (<5 nm) to overcome the charging effect»; line 138, «vacuum degree ≤ 2 × 10−7 mbar» (missing superscript), line 141, «284.8eV» (missing space), lines 168-170, «The zeta potential (ζ) on the surface of the membrane sample was measured by a zeta potential analyzer (SurPASS, Anton Paar GmbH, Austria) was used to measure the zeta potential (ζ) on the surface of the membrane sample.»; lines 179-183, «Firstly, test the membrane permeation flux of pure water at 1 bar, and then test the concentration of 7 bar concentration 500 ppm, 10 bar concentration 1500 ppm, 15.5 bar concentration 2000 ppm, 55.2 bar concentration 32000 ppm concentration 2000 ppm Flux and retention of NaCl solution.» (where the last part came from?), line 278 «the Si-C Si-O-Si» (missing comma). I might've missed some, the authors are advised to check the typesetting of entire manuscript.

Reply:

Thank you for your careful inspection. We have corrected the corresponding part in the manuscript. For example:

The original manuscript (Section 3.3, line 16): while the NaCl retention was achieved. Increase (TFC 98.5%; TFN-0.2 98.9%)

The modified manuscript: while the NaCl retention was also improved (TFC 98.5%; TFN-0.2 98.9%).

Lines 159,162 – I suggest to use the capital L for litres so they can't be mistaken for capital L. It is strange that some parts of the manuscript use capital L and some small l.

Reply:

The L in the unit has been expressed in capital letters.

Line 35, «mixed matrix membrane (TFN)», and line 100, «PA membrane (TFC and TFN)» - I guess TFC means «thin film composite» and TFN means «thin film nanocomposite», but these abbreviations were not introduced.

Reply:

As you guessed, TFN is a thin-film nanocomposite membrane. It is obtained by introducing nanoparticles into the thin-film composite membrane (TFC). This has been explained in the manuscript (Section 1, the second paragraph, line 1~3).

Section 2.2 seems to be partly written in imperative. Can't say this is frequent stylistic choice. The authors might consider rewording.

Reply:

This part of the content has been unified into the “Times New Roman” font.

Also on section 2.2: why only the concentrations and not amounts of substances are given?

Reply:

The corresponding mass volume concentration has also been added to the manuscript (Section 2.2, line 2, line 4~6).

Line 97, PSF abbreviation was not introduced.

Reply:

The PSF is the polysulfone membrane. This is marked in the manuscript (Section 2.1, line 4).

Line 120 – «for the back side of the film, transfer the separation layer to the silicon wafer,» - can authors please provide details of transfer protocol?

Reply:

Figure S 2. PA layer back transfer process.

Chloroform (CHCl3) is used to dissolve the PSF base layer (Figure S 2). Place the TFN membrane face down and float it in chloroform. After the PSF is dissolved, the non-woven fabric is separated from the PA layer. Then, we deposit the floating PA layer on a silicon wafer. The membrane surface structure on the silicon wafer is the PA back.

The above discussion has been added to the supplementary information (SI, Section 2). The corresponding content in the manuscript (Section 2.3, line 5) is marked.

Lines 168-176, regarding zeta potential protocol: besides formatting errors being fixed I would ask for a more detailed description. Did the authors use one sample or double samples? How the titration was implemented? Why NaOH and not KOH was used to adjust the pH of KCl solution?

Reply:

We use double samples. The measured result is the membrane surface flowing potential. The instrument titrates automatically during the test. The pH value was adjusted by NaOH and HCl. This is pointed out in the manuscript (Section 2.3, the seventh paragraph, line 5). Some duplicate content has also been deleted.

Line 199, «cavernous protrusions» - maybe it was intended as «cavernous depressions»?

Reply:

This is our description of the mistake. The following corrections have been made.

The original manuscript (Section 3.1, line 3): Small cavernous protrusions are arranged between the spheres.

The modified manuscript: Small cavernous depressions are arranged between the spheres.

Line 204, «the reaction rate of hydrolysis polycondensation» - this reaction is not described in text. In fact, no reactions were shown. May the authors please provide the equations for mentioned reactions?

Reply:

Figure 2. Interfacial polymerization process.

During the formation of the TFN composite layer, PTS is connected to the polyamide network by hydrolysis and substitution of amino hydrogen (Figure 2). The related content has been added to the manuscript (Section 3.1, line 7).

Figure 1 – did the authors compare SEM visualisations of modified membrane with SEM of pristine PSF substrate membrane?

Reply:

Figure S 3. SEM image of PSF membrane.

The PSF base membrane has a smooth surface without special structure (Figure S 3). In comparison, TFN membrane has a special fold structure (Figure 1 B~F), which forms a unique transfer process.

The above content has been added to the supplementary information (SI, Section 3). The related content is also marked in the manuscript (Section 3.1, line 1).

Line 218 – I don't know what junction saving is. Maybe it's just me, but the authors might consider adding the elaboration.

Reply:

This is an expression error. This is just to explain the pore size distribution (Figure 3) on the back of the PA layer. The following corrections have been made in the manuscript.

The original manuscript (Section 3.1, the second paragraph, line 2): Analyzing the hole on the back side, it can be concluded that the PA junction saving is 10-35 nm (Figure 2).

The modified manuscript: It can be concluded that the PA back pore is 10-35 nm (Figure 3).

What is the meaning of peak pore diameter (line 219)? Judging by given numbers and figured out its not a maximum or diameter. Is it mean value of pore diameters?

Reply:

This is obtained by counting the probability of pore size distribution in the picture and performing mathematical fitting. The peak pore diameter is the highest value of the fitted curve. It means that the size is the largest probability distribution in the SEM image. The value is closely related to membrane performance. The curve fitting process is shown in the Figure S 4.

Figure S 4. Pore size distribution fitting curve.

Figure 2: how the fitting curve (red) of pore diameters war built?

Reply:

The aperture fitting curve is obtained by nonlinear fitting of the gauss model. The detailed details are shown in the Figure S 4. It has been added to the supplementary information. It also has been marked in the manuscript (Section 3.1, the second paragraph, line 5).

Lines 248-249, «We speculate that the protrusion is caused by the violent heterogeneous eruption of MPD in the water phase of PSF membrane pore [26].» - was porosity of PSF membrane evaluated?

Reply:

The statement here may be unclear. The interfacial polymerization process mainly occurs between the emulsified layer at the junction of the organic phase and the water phase. The PSF membrane pores have little effect on this process. The following corrections have been made.

The original manuscript (Section 3.1, the third paragraph, line 12): We speculate that the protrusion is caused by the violent heterogeneous eruption of MPD in the water phase of PSF membrane pore.

The modified manuscript: The interfacial polymerization process mainly occurs between the emulsified layer at the junction of the organic phase and the water phase. MPD molecules close to the emulsified layer can quickly participate in the polymerization reaction. The reaction is slower for the MPD away from the emulsified layer. The difference in the reaction causes the protrusion.

Lines 332 onwards – I guess the percentages show us the increase of values in relation to membrane without the dopant? The authors may consider explicitly stating it.

Reply:

Yes. These percentages are relative to the TFC membrane without PTS. The relevant parts in the manuscript are marked (Section 3.3).

Line 350, «A slight excess of PhSiCl3» - I don't understand the use of the word «excess» here. Why this amount of dopant was considered excessive?

Reply:

This is a mistake in writing. This refers to a small amount of PTS. The following corrections have been made.

The original manuscript (Section 3.3, the third paragraph, line 5): A slight excess of PhSiCl3

The modified manuscript: A small amount of PTS

Lines 353-354, «the adjustment space of the formula is increased, which is conducive to the stable control of industrial production.» - might be elaborated further, since I didn't see the discussion of adjustment space prior to this.

Reply:

Our discussion here is unclear. This is our negligence. The following corrections have been made.

The original manuscript (Section 3.3, the third paragraph, line 8): The PhSiCl3 with polyamide is better, the adjustment space of the formula is increased, which is conducive to the stable control of industrial production.

The modified manuscript: Considering compatibility and dispersion, adding PTS to the organic phase is an effective way to improve membrane performance. The above test and analysis have demonstrated this. For low-pressure desalination process, the amount of PTS added can be adjusted to prepare high-performance TFN membranes.

Round 2

Reviewer 1 Report

The authors of the manuscript have noted my previous questions and comments. At the moment, the manuscript can be reviewed by an editor for possible publication.

A few minor notes:

Line 101. Remove "In the work"
Lines 125, 199... I couldn't find supplementary information (SI, Figure S 2), (SI, Figure S 3) ...
Line 129. I recommend replacing "shoot" for example with "scan"
Line 178. Remove "."
Line 311. I recommend adding information about the composition instead of "with different PhSiCl3 loading" to the caption. 

Author Response

Dear editor, reviewers:

Thank you for giving us another opportunity to revise the manuscript. We tried our best to answer the reviewer’s questions. The following is the detailed response.

Response to Reviewer 1 Comments

The authors of the manuscript have noted my previous questions and comments. At the moment, the manuscript can be reviewed by an editor for possible publication.

A few minor notes:

Line 101. Remove "In the work"

Reply:

It has been removed (Section 2.2, line 1).

Lines 125, 199... I couldn't find supplementary information (SI, Figure S 2), (SI, Figure S 3) ...

Reply:

The supplementary information may fail to be added to the document when uploading. Now it has been added to the end of this document.

Line 129. I recommend replacing "shoot" for example with "scan"

Reply:

The “shoot” has been replaced with “scan”.

The original manuscript (Section 2.3, line 9): Finally, place the processed sample in the chamber of the instrument and shoot at an ac-celeration voltage of 5.0-15.0 kV.

The modified manuscript: Finally, place the processed sample in the chamber of the instrument and scan at an ac-celeration voltage of 5.0-15.0 kV.

Line 178. Remove "."

Reply:

It has been removed.

Line 311. I recommend adding information about the composition instead of "with different PhSiCl3 loading" to the caption.

Reply:

We sincerely adopted this suggestion and made such a change.

The original manuscript (Section 3.2, Figure 7 caption): Figure 7. C 1s high-resolution spectra of the top surfaces for the membranes with different PhSiCl3 loading.

The modified manuscript: Figure 7. C 1s high-resolution spectra of the top surfaces for the membranes with different covalent silicon composition.

The permeability and selectivity of polyamide reverse osmosis membrane were significantly enhanced by PhSiCl3

Junjie Yu, Kaifeng Gu, Binbin Yang, Kaizhen Wang, Yong Zhou*, Congjie Gao

Center for Membrane and Water Science & Technology, Zhejiang University of Technology, Hangzhou 310014, P. R. China

*Corresponding Author: E-mail: [email protected] (Y.Z.)

  1. TEM images of TFN and TFC membranes.

 Figure S 1. TEM images of the membranes with different PTS loading: (A, a) TFC membrane; (B, b) TFN-0.1 membrane; (C, c) TFN-0.2 membrane; (D, d) TFN-0.3 membrane; (E, e) TFN-0.4 membrane; (F, f) TFN-0.5 membrane.

We can see that with the increase of the concentration of PTS, the protuberance of TFN membrane is higher and higher, and the cross-section structure is more and more clear. This can be attributed to the introduction of silicon element and cavity, which increases the contrast of cross-section TEM image.

  1. PA layer transfer method.

Figure S 2. PA layer back transfer process.

Chloroform (CHCl3) is used to dissolve the PSF base layer (Figure S 2). Place the TFN membrane face down and float it in chloroform. After the PSF is dissolved, the non-woven fabric is separated from the PA layer. Then, we deposit the floating PA layer on a silicon wafer. The membrane surface structure on the silicon wafer is the PA back.

  1. SEM image of PSF membrane.

Figure S 3. SEM image of PSF membrane.

The PSF base membrane has a smooth surface without special structure (Figure S 3). In comparison, TFN membrane has a special fold structure (Figure 1 B~F), which forms a unique transfer process.

  1. Aperture fitting.

Figure S 4. Pore size distribution fitting curve.
